# Novel Candidate *loci* and Pathogenic Germline Variants Involved in Familial Hematological Malignancies Revealed by Whole-Exome Sequencing

**DOI:** 10.3390/cancers15030944

**Published:** 2023-02-02

**Authors:** Cristina Andrés-Zayas, Julia Suárez-González, María Chicano-Lavilla, Mariana Bastos Oreiro, Gabriela Rodríguez-Macías, Patricia Font López, Santiago Osorio Prendes, Gillen Oarbeascoa Royuela, Patricia García Ramírez, Rocío Nieves Salgado, Ignacio Gómez-Centurión, Diego Carbonell Muñoz, Paula Muñiz, Mi Kwon, José Luis Díez-Martín, Ismael Buño, Carolina Martínez-Laperche

**Affiliations:** 1Genomics Unit, Gregorio Marañón General University Hospital, Gregorio Marañón Health Research Institute (IiSGM), 28009 Madrid, Spain; 2Gregorio Marañón Health Research Institute (IiSGM), 28009 Madrid, Spain; 3Department of Hematology, Gregorio Marañón General University Hospital, 28007 Madrid, Spain; 4Department of Hematology, Hospital Universitario Príncipe de Asturias, 28802 Madrid, Spain; 5Department of Hematology, Hospital Universitario Fundación Jiménez Díaz, 28040 Madrid, Spain; 6Department of Medicine, School of Medicine, Complutense University of Madrid, 28040 Madrid, Spain; 7Department of Cell Biology, School of Medicine, Complutense University of Madrid, 28040 Madrid, Spain

**Keywords:** predisposition syndromes, whole-exome sequencing, cancer susceptibility, hematological malignancies, germline mutation

## Abstract

**Simple Summary:**

Inherited predisposition to hematological malignancies is more common than previously perceived. The main objective of our study was to analyze the whole-exome sequencing data for the genomic characterization of sixteen patients with a strong family or personal onco-hematological history. When a duo analysis was performed, we detected pathogenic or likely pathogenic (P/LP) germline variants in four out of the six families studied. In the remaining four individuals, we detected three P/LP germline variants in genes with a potential role in cancer development. Next-generation sequencing strategies lead to the identification of novel candidate genes (*NFATC2* and *TC2N*) potentially involved in the development of these germline syndromes. The recognition of predisposing variants is crucial for disease management and follow-up of affected patients and their relatives.

**Abstract:**

The familial occurrence of hematological malignancies has been underappreciated. Recent studies suggest that up to 15% of adults with myeloid neoplasms carry germline pathogenic variants in cancer-predisposing genes. This study aimed to identify the underlying germline predisposition variant in patients with a strong family or personal onco-hematological history using whole exome sequencing on sixteen uncharacterized individuals. It was carried out in two groups of patients, one with samples available from two affected relatives (Cohort A) and one with available samples from the index case (Cohort B). In Cohort A, six families were characterized. Two families shared variants in genes associated with DNA damage response and involved in cancer development (*CHEK2* and *RAD54L*). Pathogenic or likely pathogenic germline variants were also found in novel candidate genes (*NFATC2* and *TC2N*). In two families, any relevant pathogenic or likely pathogenic genomic variants were identified. In Cohort B, four additional index cases were analyzed. Three of them harbor clinically relevant variants in genes with a probable role in the development of inherited forms of hematological malignancies (*GATA1*, *MSH4* and *PRF1*). Overall, whole exome sequencing is a useful approach to achieve a further characterization of these patients and their mutational spectra. Moreover, further investigations may help improve optimization for disease management of affected patients and their families.

## 1. Introduction

The significance of germline mutations in the field of solid tumors is well established, and changes in the clinical practice to include the consideration of such mutations have been settled [1,2]. However, inherited forms of hematological malignancies (IFHM) have recently been recognized, which makes it difficult to establish the precise incidence and prevalence of these disorders. While it is true that the inclusion of the category “myeloid neoplasm with germline mutations” in the World Health Organization (WHO) 2016 classification constituted a breakthrough [3], further work remains to be completed to refine the clinical diagnosis of these families and provide appropriate genetic counseling. 

Since the first gene (*RUNX1*) was identified as being responsible for the familial transmission of platelet disorders [4,5], the list of genes involved in the inheritance of hematological neoplasms has exponentially increased [6,7]. The rapid advent of next-generation sequencing (NGS) in the clinical setting has enabled the detection of germline variants in patients not otherwise diagnosed. To date, more than 20 genes have been linked with heritable hematological malignancies, several of which are also mutated in sporadic cases (*CEBPα*, *TP53*, *ETV6*) while others are restricted to unique families (*ANKRD26*, *SRP72*, *SAMD9*) [8,9,10]. Additionally, some cancer predisposition genes are not exclusive from solid tumors but are also involved in IFHM. 

In the clinical setting, detecting familial cases is crucial in the field of hematology due to the implications for the patients themselves, their family members and genetic counseling. Taking into consideration that hematopoietic stem cell transplantation (HSCT) is frequently the only curative treatment for these patients, the presence of a germline predisposing genomic variant is a significant issue in the setting of HSCT [11]. 

Relatives are at risk of sharing the same germline variant as the patient. Choosing related donors who harbor pathogenic germline variants could lead to graft failure, early relapse or donor cell leukemia, among other complications [12]. That is the reason why determining the absence of the alteration is crucial during the selection process of the proper donor. Moreover, family members should be involved in a surveillance program.

These entities are still difficult to manage for most clinicians, and there is a lack of standardization in the field regarding which patients and which genes should be tested. However, the new European LeukemiaNet recommendations give clinical features that prompt consideration of clinical testing for a germline predisposing alteration [13]. The usefulness of targeted gene panels has been largely evaluated in the literature [14,15,16], and they are considered an effective approach to detect germline variants in well-established genes at diagnosis [17]. As genomic studies provide highly sensitive information, it is crucial that all patients provide written informed consent according to the principles of the Declaration of Helsinki during a pre-test consultation. In a previous study from our group, 6% of unselected patients were confirmed to have a diagnosis of myeloid neoplasms with germline predisposition (MNGP) employing a gene panel-based approach. This percentage increased to 60% when patients with highly suggestive family histories were considered. Nevertheless, there is still a subset of patients with strong family backgrounds (two or more relatives diagnosed with onco-haematological diseases or personal background of multiple tumors) in which no mutation has been detected [12,17]. For these families, expanding the genetic analysis with more extensive technologies, such as whole exome sequencing (WES) or whole genome sequencing (WGS), would offer a unique opportunity to expand our knowledge and understand their clinical heterogeneity. 

To address this issue, the present study included a series of 16 uncharacterized patients with a strong family or personal onco-hematological history in which we performed WES with the objective to determine the underlying germline predisposition variant. For six of the patients, samples from two family members were also available for analysis. 

## 2. Patients and Methods

### 2.1. Patient Selection

A cohort of 16 human subjects was enrolled in this study, including 4 patients diagnosed with hematological malignancies and 12 relatives from 6 families. They were all recruited from the department of hematology at Hospital General Universitario Gregorio Marañón, Hospital Universitario Príncipe de Asturias and Hospital Universitario Fundación Jiménez Díaz. Patients were only included if they presented strong family backgrounds (two or more first-degree relatives affected by any malignant neoplasm) or a personal history of multiple tumors throughout their lifetime. 

The term index case was referred to the first patient referred to a medical consultation at the department of hematology.

Seven patients diagnosed with myeloid neoplasms were initially evaluated employing a gene panel implemented in the clinical setting, which included 15 genes known to be involved in MNGP: *ANKRD26*, *ASXL1*, *CBL*, *CEBPα*, *CSF3R*, *DDX41*, *ETV6*, *GATA2*, *IKZF1*, *JAK2*, *MPL*, *NF1*, *PTPN11*, *RUNX1* and *TP53*. None of the patients presented a genomic alteration in the genes analyzed. Clinical data of the patients were obtained from their medical records. 

Genomic DNA was extracted from bone marrow (BM) or peripheral blood (PB) samples according to the manufacturer’s instructions using a Maxwell^®^ RSC Blood DNA Kit (Promega, Madison, WI, USA). Across our series of 16 individuals (Table 1), our study included samples from 2 affected individuals in 6 families (Cohort A; *n* = 12 samples). This strategy allowed us to perform genomic analysis on sharing variants. In the remaining 4 families (Cohort B), material was available from the index case only (4 samples). The research protocol was approved by the Ethical Committee of University Hospital Gregorio Marañón and was conducted in accordance with the principles of the Declaration of Helsinki.

### 2.2. Genomic Profiling

A total of 16 individuals from 10 different families (Table 1) were evaluated with WES. WES libraries were prepared from 210 ng of genomic DNA using the Twist Human Core Exome Kit (Twist Bioscience, San Francisco, CA, USA) coupled with Twist Human RefSeq Panel (Twist Bioscience, San Francisco, CA, USA), according to the manufacturer’s protocol. Libraries were denatured and paired-end sequenced with a NextSeq 550 instrument (Illumina, San Diego, CA, USA) using the high output 300 cycles kit (2 × 149 bp).

### 2.3. Variant Analysis

WES data were processed using the SOPHiA DDM^®^ platform. In brief, read data were aligned to the human reference genome (GRCh37/hg19) and variants and indels were called according to the Genome Analysis Toolkit (GATK) [18]. Variants with a sequencing depth of ≤25 were filtered out. Following quality control, variants were classified as pathogenic or likely pathogenic if they met the following criteria: (1) minor allele frequency in the general population (ExAC and gnomAD databases) < 1%; (2) variant allele frequency (VAF) > 30%; (3) non-synonymous variants located in coding or splicing regions of canonical isoforms; (4) not meeting any benign criteria following ACMG guidelines [19] and not rated as a benign or likely benign variant in ClinVar database; and (5) predicted to be pathogenic by, at least, three out of four tools employed to evaluate their functional effects: Polymorphism Phenotyping v2 (PolyPhen2), MutationTaster, Sorting Tolerant from Intolerant (SIFT) and Provean. 

For the analysis, two groups were established based on the availability of samples from the index case and one affected family member (Cohort A) or from the index case only (Cohort B). To search for a disease-causing variant in Cohort B, genes highly related to cancer predisposition syndromes were first analyzed (Appendix A). If the analysis remained negative, it was expanded to whole exome sequencing data. In Cohort A, samples from affected relatives were analyzed simultaneously to retain all the variants shared by both family members.

The germline status of the variants detected was established using the VAF. When the VAF is 30–60%, the heterozygous status is very likely; meanwhile, when the VAF is >90%, the homozygous status is established.

Copy number variants (CNV) were not included in the analysis since the bioinformatic pipeline used was not designed for this purpose. 

**Table 1 cancers-15-00944-t001:** Clinical data of the patients included in this study.

Family ID	Ind.	Relationship	Age/Sex	HM	Cytogenetics	Additional Somatic Variants	Germline Variant	FH/PH	Affected Relatives	Unaffected Relatives
1	A1.1	Daughter	16/F	HL	NA	NA	*NFATC2*	Unknown	Unknown	Unknown
	A1.2	Mother ^¥^	49/F	HL	NA	NA				
2	A2.1	Sister ^¥^	65/F	MDS	46,XX(20)	NA	Not detected	AML, breast cancer	Father, two sisters, and one brother	Mother, two sisters, and one brother
	A2.2	Brother	NA	AML	NA	NA				
3	A3.1	Sister ^¥^	69/F	CLL	46,XX,t(1;18)(q12;q22)(5)/44,XX,-2,add(4)(q31), -10,-16,-17,der(18)t(1;18)(q12;q22),+mar1,+mar2(2)/46,XX(13)	NA	*TC2N*	CLL	Two sisters	Two brothers
	A3.2	Sister	70/F	CLL	46,XX(11)	NA				
4	A4.1	Sister ^¥^	NA/F	AML	NA	NA	*CHEK2*	Gallbladder cancer, lung cancer, AML	Father and two brothers	Mother and one unaffected sister
	A4.2	Brother	60/M	AML	NA	NA				
5	A5.1	Sister	NA	MDS	46,XX,inv(9)(p12q13)(20)	NA	*RAD54L*	Breast cancer, LMS, MDS, lung cancer	Two brothers and three sisters	Two brothers and two nephews
	A5.2	Brother ^¥^	NA	MDS	46,XY(20)	*CSF3R, DNMT3A, TET2, ASXL1, RUNX1, ZRSR2*				
6	A6.1	Son	46/M	MDS	46,XY(7)	Not detected	Not detected	Unknown	Unknown	Unknown
	A6.2	Father ^¥^	86/M	AML	45,XY,t(3;21)(q13; q22),-7(17)/46,XY(3)	*SRSF2, SETBP1*				
	B1		57/F	PV, AML	45,XX,-7(10)	*JAK2, RUNX1, ASXL1, KRAS*	*PRF1*	MDS uterine cancer (personal)	Mother	Four brothers and one daughter
	B2		69/F	AML	46,XX,del(5)(q13q33)(1)/46,XX(19)	*SF3B1, RUNX1*	Not detected	Leukemia, AML, PV	Grandfather, one brother, and one niece	Unknown
	B3		29/M	AML	46,XY(20)	*MPL*	*GATA1*	ALL (personal), GUS	Mother	Unknown
	B4		45	ET	NA	Not detected	*MSH4*	PV, ET	Mother and father	Unknown

Abbreviations: F: female, M: male, Ind.: individual, HM: hematological malignancies, NA: not available, HL: Hodgkin’s lymphoma, MDS: myelodysplastic syndrome, AML: acute myeloid leukemia, NA: not available, CLL: chronic lymphocytic leukemia, PV: polycythemia vera, ET: essential thrombocythemia, GUS: gammopathy of uncertain significance, FH/PH: family/personal history of hematological and/or solid malignancies, LMS: leiomyosarcoma. The index case in each family was represented by ^¥^.

### 2.4. Variant Validation

To eliminate possible artefactual and platform-specific variants, 7 selected variants identified with WES in 7 candidate genes were further validated with Sanger sequencing. The confirmation was performed by optimizing custom PCR assays for each variant followed by Sanger sequencing with Big Dye Terminator v3.1 Chemistry using an ABI Prism 3730xl Genetic Analyzer (Applied Biosystems, Foster City, CA, USA). Visualization and localization of variants were assessed by using Chromas Software and the Basic Local Alignment Search Tool (BLAST). To confirm the homozygous state of the variant found in *CHEK2*, the analysis of the samples was performed with the SALSA MLPA Probemix P190 CHEK2 kit (MRC-Holland, Amsterdam, The Netherlands) which contains probes for all 15 exons of the gene (NM_007194.4). The samples were loaded onto a 3730xl Genetic Analyzer (Applied Biosystems, Foster City, CA, USA). Results were visualized using Coffalyser.net software (MRC-Holland, Amsterdam, The Netherlands).

### 2.5. SNP-Array

As the panel employed did not capture structural abnormalities, an SNP array was performed on those families where the variant analysis remained negative. DNA samples were hybridized to the CytoSNP-12v2.1 array (Illumina, San Diego, CA, USA) according to the manufacturer’s protocol. Copy number analysis was performed using BlueFuse Multi software.

## 3. Results

### 3.1. Variant Analysis

The average number of reads per sample was 57,576,203 (range: 48,642,142–74,957,334) and 88% of reads had a minimum depth coverage of 50×. Among the altered genes, seven variants in seven strong candidates with relevant oncogenic potential were classified as pathogenic or likely pathogenic (Figure 1). 

### 3.2. Cohort A: Families with Two Affected Members Studied

Exomes from 12 samples (6 index cases and 6 affected family members) were sequenced. Among the families studied, two of them (Family 2 and Family 6) did not show any relevant genomic variant in cancer-related genes. 

Both patients in Family 2 were diagnosed with myeloid neoplasms (MDS and AML). Additionally, their father had AML and their sister had breast cancer. On the other hand, the pedigree of Family 6 was not available. We only had the information regarding the index case (diagnosed with MDS) and his father (diagnosed with AML).

The remaining four families shared variants between the affected members in genes potentially involved in cancer development: *NFATC2* (c.1101-1G>A, p.(?)), *TC2N* (c.949C>T, p.Arg443*), *CHEK2* (c.478A>G, p.Arg160Gly), and *RAD54L* (c.863del, p.Gly288Glufs*28) (Table 1 and Table 2). 

The heterozygous germline *NFATC2* variant was present in the index case and her mother from Family 1, both diagnosed with Hodgkin’s lymphoma at 46 and 16 years old, respectively. The functional relevance of the intronic variant (c.1101-1G>A) remains uncertain, but it affects the natural donor-acceptor splice site and it is absent in population databases. 

In Family 3, the index case and her sister, who had all presented with chronic lymphocytic leukemia (CLL), showed a novel truncating heterozygous variant in the *TC2N* gene. 

Both patients in Family 4 harbored a homozygous variant (c.478A>G) in the *CHEK2* gene. The index case in this family was initially found to have acute myeloid leukemia (AML). Her older brother had lung cancer at 63 years of age. Her younger brother presented with AML at 60 years of age, and he received an allo-HSCT from her unaffected sister and died from donor cell leukemia. Donor cell leukemia is a rare, but severe, complication post-transplant. The development of these malignancies occurs in donor cells. Her father had gallbladder cancer and on the maternal side multiple family members presented with solid tumors. 

In Family 5, both affected individuals were diagnosed with myelodysplastic syndrome (MDS) at 75 years. The variant reported herein (c.863del) in the *RAD54L* gene leads to a stop codon, generating a smaller non-functional protein. Highly conserved amino acids are affected, and critical motifs are eliminated. Moreover, this family presented a strong family background of other solid tumors, a feature frequently associated with *RAD54L* variants.

### 3.3. Cohort B: Families with Only the Index Case Studied

The additional four index cases harbor three variants in genes with a probable role in the development of IFHM (Table 1 and Table 3). 

Patient B1 was diagnosed with AML at 57 years of age. Her family history included a mother with AML. The genetic analysis identified a likely germline variant in the *PRF1* gene (c.272C>T). Perforin plays a central role in the pathophysiology of the immune system and is essential for eliminating cancerous cells. 

Patient B2 was also diagnosed with AML at 69 years of age. Her family background is strong including a brother diagnosed with biphenotypic acute leukemia, a grandfather diagnosed with “some sort of leukemia” and a niece diagnosed with polycythemia vera (PV). The WES analysis did not show any potentially pathogenic germline alterations. 

Patient B3 did not show any relevant family history, but he had presented acute lymphoblastic leukemia (ALL) during childhood and several years later developed an AML. Among the altered genes, a likely germline variant in *GATA1* (c.-19-679_221-48delinsTC) was retained as a strong candidate that conferred germline risk for hematological malignances or solid tumors. This mutation impaired the production of a full-length form of the protein.

Patient B4 presented essential thrombocytemia (ET) at 45 years of age, and her medical records showed that both parents had suffered from myeloproliferative neoplasms (mother with ET and father with PV). The likely germline variant found in this case (*MSH4*:c.56C>A) generates a stop codon in a member of the DNA mismatch repair mutS gene family [20]. 

### 3.4. SNP-Array

Regarding the analysis of CNV, it did not reveal any germinal chromosomal alterations in the patients studied with SNP-array (both members of Family 2, Family 6, and patient B2). Somatic monosomy 7 (-7) was detected in one of the patients, as it had also been observed using conventional cytogenetics at diagnosis. 

The homozygous state of the variant detected in the *CHEK2* gene suggested a possible deletion of the other allele. However, MLPA did not reveal any CNV alteration in this gene. To date, the functional role of a homozygous variant in *CHEK2* remains uncertain.

## 4. Discussion

The introduction of NGS platforms into the diagnostic setting has refined the diagnosis, prognosis and treatment of sporadic hematological malignancies [16]. Moreover, these approaches have also revealed germline variants with clinical significance in these patients [21]. 

Unlike in solid tumors, the familial occurrence of hematological malignancies has been underappreciated. Recent studies suggest that up to 15% of adults with myeloid neoplasms carry germline pathogenic variants in cancer-predisposing genes [22].

In our study, possibly contributing mutations in *NFATC2* and *TC2N* genes have been identified. On the one hand, *NFATC2* is a member of the NFAT gene family and encodes for a protein involved in cell cycle arrest, apoptosis, and inhibition of the Stat5 pathway [23]. On the other hand, the protein encoded by *TC2N* has crucial roles in cellular metabolism, proliferation and cancer [24,25]. To our knowledge, variants in those genes have not been reported previously in the literature. Their biological functions are highly related to cancer development, highlighting their candidacy as novel cancer-predisposing genes. Both alterations occurred in families with diseases affecting lymphocyte proliferation (lymphoma and CLL). These diseases are less studied in the context of predisposing syndromes and germline variants than myeloid malignancies, but their pathogenesis has a strong association with immune evasion mechanisms.

All the patients described herein and diagnosed with myeloid neoplasms are selected cases in which we had failed to detect variants in the genes commonly mutated in familial myeloid neoplasms (such as *CEBPA, RUNX1, GATA2, DDX41, ANKRD26*, etc.) [17].

*CHEK2* is considered a tumor suppressor gene involved in the regulation of cell division and the response to DNA damage [26]. Inherited germline variants have been associated with multiple cancer predisposition syndromes: many types of solid tumors, Li-Fraumeni syndrome or lymphoid and myeloid malignancies [27,28,29,30]. 

*MSH4* belongs to a member of the DNA mismatch repair (MMR) mutS family. It is involved in the process of DNA damage response and promotes genomic stability. Other members of the MMR gene family, such as *MSH2, MLH1, MSH6* and *PMS2*, are linked to Lynch syndrome with an autosomal dominant inheritance pattern [31,32]. Even though *MSH4* has not been identified in Lynch syndrome patients, its similar biological function makes it very likely to produce an effect on the genetic susceptibility to cancer. 

The *RAD54L* gene plays an important role in the DNA repair pathways [33]. Dysregulation of these mechanisms alters genomic stability and ultimately contributes to increasing the risk of developing multiple types of cancer [34]. 

Regarding the *GATA1* gene, it has a crucial role in normal human hematopoiesis. Acquired and inherited mutations in *GATA1* contribute to Diamond–Blackfan anemia, acute megakaryoblastic leukemia, transient myeloproliferative disorder and a group of related congenital dyserythropoietic anemias with thrombocytopenia [35,36].

The *PRF1* gene has been historically associated with familial hemophagocytic lymphohystiocytosis-2 with an autosomal recessive inheritance pattern [37]. However, more recently, inherited *PRF1* mutations were subsequently described in various types of lymphomas, suggesting an involvement in the immune surveillance mechanisms preventing tumor growth and/or development. Escape from immune surveillance is thought to be the main mechanism possibly explaining the role of *PRF1* genetic mutations in the development of leukemia and lymphoma [38,39]. Regarding the variant found in Pt. B1 (p.Ala91Val), it has been previously reported by other authors. This substitution in exon 2 was first described as a neutral polymorphism. Nonetheless, recent clinical evidence suggests a potential pathogenic role for the variant. Functional assays revealed low perforin expression levels, as well as impaired NK cell-mediated cytotoxicity. It has been identified in patients with hematological malignancies, lymphoproliferative diseases and aplastic anemia [40,41,42,43]. Analogous to the cases reported, our findings also support the idea that this perforin mutation can contribute to hereditary cancer predisposition. However, another somatic event seems to be necessary to produce clinically significant effects and may act as a synergistic factor with other genetic mutations predisposing patients to a larger range of cancer types [44]. 

No genomic variants met our selection criteria in two families (Families 2 and 6) and one index case (Pt. B4). These examples highlight perfectly the challenges faced by researchers in identifying predisposing genomic lesions. It is conceivable that such families represent good candidates for WGS as the variant may reside outside the coding region. 

Recurrent gene alterations have not been detected in the patients analyzed, suggesting that most families are linked to a particular variant with its own characteristics regarding expressivity, penetrance and clinical heterogeneity. Efforts must be made to define the precise contribution of novel cancer-related genes and provide appropriate genetic counseling to patients and at-risk family members. Identification of patients with hematological malignancies with germline predisposition is crucial in the allogeneic HSCT setting since it dictates the selection of donors since relatives with the mutation would not be considered. 

In our experience, when samples were available from two affected individuals, candidate variants were detected in four out of the six families analyzed, which represent 66% of the cases. However, when samples were available from the index case, the percentage of patients with a candidate germline variant was 75%. These percentages are relatively similar, so according to our results, it is feasible to perform WES even in cases where a sample is available from the index case only. It is worth noting that this approach leads to the detection of a suggestive candidate variant. Nevertheless, in the absence of a proper segregation analysis, the possibility that the genomic finding does not have a causative role cannot be ruled out. 

Additionally, efforts must be made to create pre-test consultations in the health system which could provide an appropriate evaluation of patients. These units would be responsible for taking into account some recommendations from expert panels [22,45]: (1) family history of AML or other hematological malignancies; (2) diagnosis of MDS at a young age (<40 years old) or a personal history of more than two hematological malignancies; (3) aberrations in chromosome 7, overall when the age of diagnosis is <50 years old; (4) long-term cytopenia or bleeding; (5) other phenotypic features related to predisposition syndromes; (6) excessive toxicity to chemotherapy agents or poor mobilization; and (7) identification of a variant in a gene related to predisposition syndromes and a VAF suggestive of germline origin (>30%). 

Based on our experience, patients who meet one or more of these criteria should be suspected of having an underlying predisposition syndrome, and WES analysis should be offered. However, if not possible, a gene panel-based approach aimed at studying a set of genes associated with cancer predisposing syndromes should be recommended.

Ideally, segregation analysis must be performed to rule out the involvement of the variant in the disease. Nevertheless, in retrospective studies, material from multiple affected individuals is not always available for several reasons such as post-decease evaluation, travel constraints or loss of follow-up. 

Furthermore, cultured skin fibroblasts offer the gold standard approach to discriminate germline from somatic changes [12]. This evaluation was not practical in our cases, so BM or PB samples obtained at complete remission were analyzed. Interpretation of the results proceeded under the recognition of this limitation, and the existence of confounding factors, including clonal hematopoiesis, cannot be ruled out. When the analysis is performed on samples from two affected individuals, the identification of a pathogenic/likely pathogenic variant is sufficient to establish the genetic basis of the disease in this specific family.

In summary, our study leads to the identification of novel deleterious cancer-predisposing mutations. These findings contribute to ongoing investigations and to the further characterization of these patients and their mutational spectra.

Expanding the analysis to WES in highly suspicious patients for a predisposing syndrome based on personal/family background is a useful strategy to detect germline variants with clinical significance and propose novel candidate genes. Moreover, it appeared promising to provide novel insights into the etiology of these complex and heterogeneous syndromes. In the clinical setting, the recognition of these predisposition syndromes is crucial for (1) the proper management of the patients, (2) choosing a suitable donor when an allo-HSCT is considered and (3) offering an appropriate genetic counseling of affected individuals and their relatives.

## 5. Conclusions

Whole-exome sequencing is a useful approach to decipher the genomic complexity of germline predisposition syndromes to hematological malignancies. Further studies in this field would improve the clinical management of affected patients and their at-risk relatives. 

## Figures and Tables

**Figure 1 cancers-15-00944-f001:**
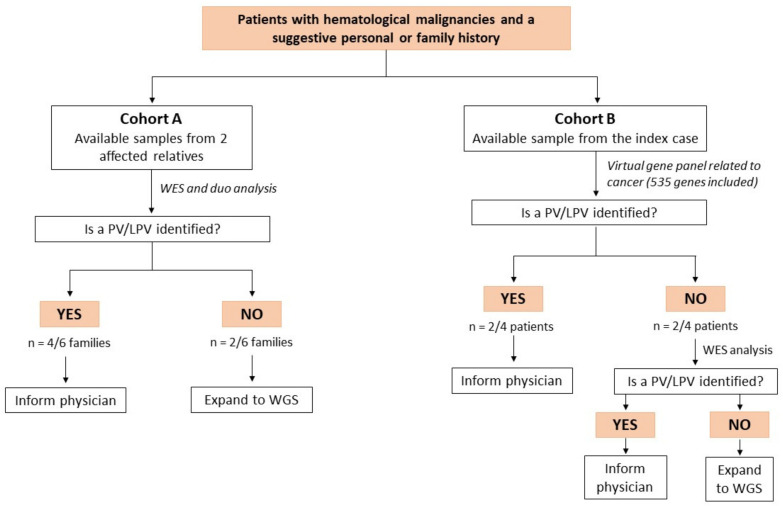
Algorithm for the detection of germline variants predisposing to inherited forms of hematological malignancies. Abbreviations: PV: pathogenic variant, LPV: likely pathogenic variant, WGS: whole genome sequencing.

**Table 2 cancers-15-00944-t002:** Genes mutated in six families from Cohort A.

Family ID	Gene	Variant	Consequence	VAF (%)	Transcript	ExAC Frequency	ACMG Status	FH/PH
1	*NFATC2*	c.1101-1G>A; p.(?)	splice acceptor	47.7	NM_001136021	No	Pathogenic	No
2	Not detected							Yes
3	*TC2N*	c.1327C>T; p.(Arg443*)	frameshift	40	NM_001128595	0	Likely pathogenic	No
4	*CHEK2*	c.478A>G; p.(Arg160Gly)	missense	100	NM_001005735	0.00018	Pathogenic	Yes
5	*RAD54L*	c.863del; p.(Gly288Glufs*28)	framsehift	42.4	NM_001142548	No	Likely pathogenic	Yes
6	Not detected							No

Abbreviations: VAF: variant allele frequency, FH/PH: family/personal history of hematological and/or solid malignancies.

**Table 3 cancers-15-00944-t003:** Genes mutated in four index cases from Cohort B.

Individual	Gene	Variant	Consequence	VAF (%)	Transcript	ExAC Frequency	ACMG Status	FH/PH
Pt. B1	*PRF1*	c.272C>T; p.(Ala91Val)	missense	51.5	NM_001083116	0.001	Likely pathogenic	Yes
Pt. B2	Not detected							Yes
Pt. B3	*GATA1*	c.-19-679_221-48delinsTC; p.(?)	no-start	94.8	NM_002049	No	Pathogenic	Yes
Pt. B4	*MSH4*	c.56C>A; p.(Ser19*)	nonsense	48.1	NM_002440	0.000004	Likely pathogenic	Yes

Abbreviations: VAF: variant allele frequency, FH/PH: family/personal history of hematological and/or solid malignancies.

## Data Availability

The data presented in this study are available in this article (and Appendix A).

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
