# Peer review of "Novel Candidate *loci* and Pathogenic Germline Variants Involved in Familial Hematological Malignancies Revealed by Whole-Exome Sequencing"

_cancers, 2023, doi:10.3390/cancers15030944_

Round 1
Reviewer 1 Report
This is an interesting investigation. The purpose of the study was to identify germline predisposition variants in patients with strong family or personal histories of hematological malignancies. The authors concluded that their findings contributed to the characterization of mutational spectra of the patients in the study. They predict that further investigations may help to optimize for disease management.
The main limitation of the study was the low number of available cases. Any statistical assumptions in this regard are not strong. Anecdotally, the study does demonstrate the potential for more detailed genetic analysis in hematological malignancies. My comments are listed below.
Abstract
pg. 1, lines 30-31: The intended meaning of this sentence (starting with “In two variants…”) is unclear. Were no variants identified?
Introduction
pg. 2, line 64: Are not germline variants inherited by definition? It may be better to state that relatives are at risk of sharing germline variants.
pg. 2, line 77: Please include mention of patient data privacy and safety protection.
Methods
pg. 3, lines 100-101: This sentence is a double negative suggesting that all of the patients had a least one alteration.
pg. 3, line 107: It is unclear at what stage (WES or WGS) variants were or were not detected in cohort A and if WGS was used for any patients in cohort B.
pg. 3, line 111: How was the germline status of the alterations established? Was allele frequency used? Were the genes listed in the Supplementary Table 1 all considered germline variants?
pg. 3, line 134: What is the source for the compilation of genes in Supplementary Table 1?
Results
pg. 4, line 162: Is this data referring to the supplementary figure flow chart? Supplementary Figure 1 might be included in the main text as it nicely summarizes the study.
pg. 4, line 183: It is unclear which sister served as donor for the allogeneic transplant in family 4. It might be helpful to provide a brief explanation of donor cell leukemia.
pg. 4, line 192: There is no text describing family 6.
pg. 10, lines 226-228: This sentence is confusing. It is another double negative (did not reveal…in none” = did reveal in all). As written, the statement suggests the opposite of what is intended.
Discussion
pg. 10, line 254: Were not RUNX and GATA detected in cohort B?
pg. 11, line 293: “Recurrent gene alterations” would be better wording. Also see the double negative comments above.
pg. 11, line 315-318: Would buccal cell samples provide sufficient material for confirmation if an alteration was found?
Author Response
Response to Reviewer 1 Comments
This is an interesting investigation. The purpose of the study was to identify germline predisposition variants in patients with strong family or personal histories of hematological malignancies. The authors concluded that their findings contributed to the characterization of mutational spectra of the patients in the study. They predict that further investigations may help to optimize for disease management.
The main limitation of the study was the low number of available cases. Any statistical assumptions in this regard are not strong. Anecdotally, the study does demonstrate the potential for more detailed genetic analysis in hematological malignancies. My comments are listed below.
Abstract
Point 1: pg. 1, lines 30-31: The intended meaning of this sentence (starting with “In two variants…”) is unclear. Were no variants identified?
Response to point 1: According to your observation, we have reformulated the sentence. Lines 45-46.
Introduction
Point 2: pg. 2, line 64: Are not germline variants inherited by definition? It may be better to state that relatives are at risk of sharing germline variants.
Response to point 2: Thank you for your comment, we rephrased the paragraph in line 80.
Point 3: pg. 2, line 77: Please include mention of patient data privacy and safety protection.
Response to point 3: According to your observation, we have clarified this aspect in the text. As you can see, we have added a new sentence in the introduction. Lines 93-95.
Methods
Point 4: pg. 3, lines 100-101: This sentence is a double negative suggesting that all of the patients had a least one alteration.
Response to point 4: Thank you so much for your observation. We have reformulated the sentence in the Methods Section. Lines 127-128.
Point 5: pg. 3, line 107: It is unclear at what stage (WES or WGS) variants were or were not detected in cohort A and if WGS was used for any patients in cohort B.
Response to point 5: WES was used for both cohorts. The difference among them was the kind of analysis employed. In cohort A, we had samples from two affected individuals so a duo approach could be performed. On the other hand, patients in cohort B were analyzed case by case (single analysis).
Point 6: pg. 3, line 111: How was the germline status of the alterations established? Was allele frequency used? Were the genes listed in the Supplementary Table 1 all considered germline variants?
Response to point 6: Due to your suggestion, we have clarified this aspect in the text. As you can see in the Methods section, we have added a new paragraph. Lines 166-168.
Point 7: pg. 3, line 134: What is the source for the compilation of genes in Supplementary Table 1?
Response to point 7: We have used several sources of information to establish the set of genes related to cancer, with well-known functions and previously reported by other authors. In particular, the consulted sources were the following: Pubmed, Genomics England PanelApp, Human Phenoype Ontology and OMIM. However, the selection was made based on the knowledge to date. We keep in mind that this dataset is flexible and should be dynamic over time. We consider that this information may not be included in the text. However, we are open to add it if you consider it necessary.
Results
Point 8: pg. 4, line 162: Is this data referring to the supplementary figure flow chart? Supplementary Figure 1 might be included in the main text as it nicely summarizes the study.
Response to point 8: We agree with you that the Supplementary Figure 1 must be included in the text as a summarize of the study. Therefore, we have made some changes in page 5 and 6 of the manuscript.
Point 9: pg. 4, line 183: It is unclear which sister served as donor for the allogeneic transplant in family 4. It might be helpful to provide a brief explanation of donor cell leukemia.
Response to point 9: Thank you for your comment. We have clarified the information regarding the sister in line 224 and in Table 1 by adding “unaffected sister”.
On the other hand, we have clarified the aspect of donor cell leukemia in the text by adding a new sentence in lines 225-226 .
Point 10: pg. 4, line 192: There is no text describing family 6.
Response to point 10: Thank you very much for your comment. As suggested, we have included a brief paragraph describing family 2 and 6. Lines 206-209.
Point 11: pg. 10, lines 226-228: This sentence is confusing. It is another double negative (did not reveal…in none” = did reveal in all). As written, the statement suggests the opposite of what is intended.
Response to point 11: Thank you so much for your observation. We have reformulated the sentence in the Results Section. Lines 269-270.
Discussion
Point 12: pg. 10, line 254: Were not RUNX and GATA detected in cohort B?
Response to point 12: Thank you so much for your observation. In the clinical setting, a gene panel based approach was performed in all the patients included in the present study. This panel includes 15 genes known to be involved in myeloid neoplasms with germline predisposition. Among these genes, GATA2 and RUNX1 are included. As explained in the third page (lines 124-127) any of the patients presented pathogenic variants in those genes.
Point 13: pg. 11, line 293: “Recurrent gene alterations” would be better wording. Also see the double negative comments above.
Response to point 13: According to your observation, the paragraph was reformulated. Lines 337-339.
Point 14: pg. 11, line 315-318: Would buccal cell samples provide sufficient material for confirmation if an alteration was found?
Response to point 14: Thank you for your observation. Eventhough, buccal cell samples could be a potential source of DNA in these patients (once they have achieved a complete remission status in order to avoid PB contamination at diagnosis, as reported in the literature), in our experience we have not achieved sufficiente material with enough quality to perform PCR and Sanger sequencing on these samples. Skin fibroblasts remain as the gold standard to confirm the germline status in onco-hematological patients.

Reviewer 2 Report
The same group of researchers published a paper at Mol Oncol in 2021 (ref #17), using targeted NSG to detect germline variants in patients with myeloid malignancies. In this submitted manuscript, the authors tried to identify germline mutations in patients with familiar blood cancers by WES. Overall, the patient sequencing data provided some new genetic information on familiar blood malignancies. The manuscript is well written, however, some additional comments should be given to better explain the experimental design and results.
1. What’s the abbreviation of MNGP (line 77)?
2. What is the control/normal sample used for WES? Why not include family members without diseases?
3. For patient selection, 16 human subjects were enrolled in this study, including 10 patents and 6 family members. What is the relational of dividing people into cohort A and B?
In stead saying 6 family members, will it be clearer by saying 12 family members from 6 families?
4. In line 167, 12 samples have 6 index cases and 6 affected family members. What are index cases?
5. In line 98-101, the patients do not carry alternations in the 15 well known genes which are commonly mutated in myeloid neoplasms, this will exclude many patient candidates? Why not include patients carrying those gene mutations? It won’t affect the results, right? since the purpose of the paper is to identify some “new” variants.
6. The authors said the mutations identified by WES are germline mutation? How to identify whether they are germline or somatic mutations? The patients involved are family members (some are elderly patients) but not necessarily share the same genetics.
7. The patients include CLL patients (group 3), so the data is not restricted to myeloid malignancies? Will the results change if only include patients with myeloid malignancies?
8. How to confirm that those variants play functional roles in familiar hematological malignances?
Author Response
Response to Reviewer 2 Comments
The same group of researchers published a paper at Mol Oncol in 2021 (ref #17), using targeted NSG to detect germline variants in patients with myeloid malignancies. In this submitted manuscript, the authors tried to identify germline mutations in patients with familiar blood cancers by WES. Overall, the patient sequencing data provided some new genetic information on familiar blood malignancies. The manuscript is well written, however, some additional comments should be given to better explain the experimental design and results.
Point 1: What’s the abbreviation of MNGP (line 77)?
Response to point 1: Abbreviation was added in the text. Line 96.
Point 2: What is the control/normal sample used for WES? Why not include family members without diseases?
Response to point 2: As part of a separate study, we have performed WES sequencing in 25 patients diagnosed with other diseases not related to oncological processes. We have also analysed the subset of genes in Supplementary Table 1 (highly related to predisposition to different tumors) in order to determine the frequency of pathogenic or likely pathogenic variants in these patients. We have detected variants of uncertain significance in all cases, but no other significant alterations. We keep in mind that this study included few normal samples but it could give us some idea of the higher frequency of detection of pathogenic or likely pathogenic variants when the patients were selected according to family/personal history of onco-hematological diseases.
In our opinion, unaffected family members may not be the best option to discard some genetic variant due to the lack of knowledge in terms of expressivity and penetrance of these syndromes. Some of the have variable or unknown expressivity and penetrance, so an unaffected relative today could be affected the day after tomorrow.
Point 3: For patient selection, 16 human subjects were enrolled in this study, including 10 patents and 6 family members. What is the relational of dividing people into cohort A and B?
In stead saying 6 family members, will it be clearer by saying 12 family members from 6 families?
Response to point 3:
Thank you very much for your comment. As suggested, we rephrased the sentence in Patients and Methods section Line 111-114.
The rational of dividing patients into two cohorts was the different way for the analysis. In cohort A, we had samples from two affected individuals so a duo approach could be performed. On the other hand, patients in cohort B were analyzed case by case (single analysis).
Point 4: In line 167, 12 samples have 6 index cases and 6 affected family members. What are index cases?
Response to point 4: Thank you for your observation. We added a sign in Table 1 to clarify our data and marked who is the index case in each family. Moreover, a new paragraph was added in the Patients and Methods section to clarify this aspect. Lines 122-123.
Point 5: In line 98-101, the patients do not carry alternations in the 15 well known genes which are commonly mutated in myeloid neoplasms, this will exclude many patient candidates? Why not include patients carrying those gene mutations? It won’t affect the results, right? Since the purpose of the paper is to identify some “new” variants.
Response to point 5: Thank you very much for your comment. We have not included patients carrying germline mutations in well known genes because clinically they already had a diagnosis and a defined entity. The main objective of this study was to identify novel variants in undiagnosed patients with strong family or presonal history of onco-hematological malignancies.
Point 6: The authors said the mutations identified by WES are germline mutation? How to identify whether they are germline or somatic mutations? The patients involved are family members (some are elderly patients) but not necessarily share the same genetics.
Response to point 6: We have clarified how a germline mutation was identified by WES adding a new paragraph in the Patients and Methods section (lines 166-168). In the cases where two relatives shared the same rare variant, the family bond is taken over. In family 6, Short Tandem Repeat analysis wasa performed to compare allele repeats at specific loci in DNA between both samples and confirm their relationship.
Point 7: The patients include CLL patients (group 3), so the data is not restricted to myeloid malignancies? Will the results change if only include patients with myeloid malignancies?
Response to point 7: Thank you very much for your observation. As these entities occur at relatively low frequencies, our study included patients with strong familiy/personal history but with a diagnosis of any hematological neoplasm. That is why we did not want to restrict the selection of cases only to myeloid neoplasms in order to get a more complete vision of this field.
Point 8: How to confirm that those variants play functional roles in familiar hematological malignances?
Response to point 8: We totally agree with your comment. There is a need to perform further functional validation studies (such as in vitro models, cell cultures or even animal models) to determine the exact impact of these variants. However, it must be performed in new studies and working tohether with other colleagues because it is not our field of expertise.
Reviewer 3 Report
Andres-Zayas and colleagues have performed whole exome sequencing on a small series (10 index cases and 6 family members) of patients deemed to be at high risk for inherited risk alleles. They identify novel risk alleles in most patients. The gene targets and their potential as drivers of disease is discussed. This is an outstanding example of discovery research performed in a clinical setting, and the authors should be commended.
Specific Criticisms
First sentence of abstract, “Unlike in solid tumors,” is generally true, but does not propel the manuscript and should be deleted for style.
The final sentences of the abstract should be reconsidered. “Overall, our findings contribute..” is not a very informative sentence, and the final sentence of the abstract should be more specific.
This criticism applies to the end of the discussion as well. What concrete advice do the authors have for clinicians regarding the findings here. It might be argued that uncovering germline predisposition does not change clinical management so what is the point?
Certainly, long term followup of family members might be warranted once germline alleles are identified. Are the authors doing this? How is follow up done? Which patients should have WES? As the authors are no doubt aware, most payors still do not support routine WES for patients. How are allo transplant donors screening in Europe? Donor selection is relevant clinical issue that might be mentioned.
In short, the identification of specific alleles in these families is interesting and expands our list of genes containing inherited risk alleles. However, in the opinion of this reviewer, the manuscript would be more valuable if the views of these experts and their clinical colleagues on how clinical care or workflow might change and adapt to a genetic landscape that is getting more and more complex. The final words of the Discussion should be an insight into how WES should be used to guide clinicians in the future.
Author Response
Response to Reviewer 3 Comments
Andres-Zayas and colleagues have performed whole exome sequencing on a small series (10 index cases and 6 family members) of patients deemed to be at high risk for inherited risk alleles. They identify novel risk alleles in most patients. The gene targets and their potential as drivers of disease is discussed. This is an outstanding example of discovery research performed in a clinical setting, and the authors should be commended.
Specific Criticisms:
Point 1: First sentence of abstract, “Unlike in solid tumors,” is generally true, but does not propel the manuscript and should be deleted for style.
Response to point 1: As suggested, we have removed it.
Point 2: The final sentences of the abstract should be reconsidered. “Overall, our findings contribute..” is not a very informative sentence, and the final sentence of the abstract should be more specific.
Response to point 2: We agree with your advice about the final sentence of the abstract and we have reformulated it. Line 49-50.
Point 3: This criticism applies to the end of the discussion as well. What concrete advice do the authors have for clinicians regarding the findings here. It might be argued that uncovering germline predisposition does not change clinical management so what is the point?
Response to point 3: Thank you so much for your comment. We rephrased the paragraph in the Discussion section. Lines 389-392.
Point 4: Certainly, long term followup of family members might be warranted once germline alleles are identified. Are the authors doing this? How is follow up done? Which patients should have WES? As the authors are no doubt aware, most payors still do not support routine WES for patients. How are allo transplant donors screening in Europe? Donor selection is relevant clinical issue that might be mentioned.
Response to point 4: In the context of our hospital, we are setting up a multidisciplinary genetic counselling unit to provide patients a proper transmission of these sensitive information and provide the rest of relatives the chance of performing the genetic analysis, if desired. However, health care systems could not afford, to date, routine WES analysis for patients.
In the setting of transplant, our centre tries to look for an unrelated donor versus a family donor when a patient has strong family history of onco-hematological disorders even though a pathogenic or likely pathogenic variant has not been detected during the routine analysis. This strategy is performed in order to avoid risks such as poor mobilization, development of donor cell leukemia, inter alia.
Point 5: In short, the identification of specific alleles in these families is interesting and expands our list of genes containing inherited risk alleles. However, in the opinion of this reviewer, the manuscript would be more valuable if the views of these experts and their clinical colleagues on how clinical care or workflow might change and adapt to a genetic landscape that is getting more and more complex. The final words of the Discussion should be an insight into how WES should be used to guide clinicians in the future.
Response to point 5:
In line with the reviewer’s comment in point 4, we added in the text a new paragraph clarifying this aspect and giving a workflow about how to proceed in accordance to expert guidelines and our own experience Lines 356-368. However, with the advent of novel works the knowledge in the field will make great strides.
Ideally, if health systems support it, pre-test consultations should be set up to assess each patient on a case-by-case basis and whether they meet the criteria for these studies, either with whole exome sequencing or with virtual panels made up of genes related to cancer predisposition syndromes as a first approach.

Round 2
Reviewer 2 Report
The quality of the revised manuscript improved by addressing some of reviewers’ comments.